# Nurturing a Respectful Connection: Exploring the Relationship between University Educators and Students in a Spanish Veterinary Faculty

**DOI:** 10.3390/vetsci10090538

**Published:** 2023-08-24

**Authors:** Ana S. Ramírez, José Raduan Jaber, Rubén S. Rosales, Magnolia Conde-Felipe, Francisco Rodríguez, Juan Alberto Corbera, Alejandro Suárez-Pérez, Mario Encinoso, Ana Muniesa

**Affiliations:** 1Faculty of Veterinary Medicine, University of Las Palmas de Gran Canaria, Trasmontaña, 35413 Arucas, Las Palmas, Spain; anasofia.ramirez@ulpgc.es (A.S.R.); ruben.rosales@ulpgc.es (R.S.R.); magnolia.conde@ulpgc.es (M.C.-F.); francisco.guisado@ulpgc.es (F.R.); juan.corbera@ulpgc.es (J.A.C.); alejandro.suarezperez@ulpgc.es (A.S.-P.); 2VETFUN, Educational Innovation Group, University of Las Palmas de Gran Canaria, Trasmontaña, 35413 Arucas, Las Palmas, Spain; 3Veterinary Clinical Hospital, Faculty of Veterinary Medicine, University of Las Palmas de Gran Canaria, Trasmontaña, 35413 Arucas, Las Palmas, Spain; mencinoso@gmail.com; 4Faculty of Veterinary, University of Zaragoza-CITA, Miguel Servet, 177, 50013 Zaragoza, Zaragoza, Spain; animuni@unizar.es

**Keywords:** respect, teacher–student relationship, veterinary students

## Abstract

**Simple Summary:**

Though knowledge and the communication capacity of teachers play a crucial role in the student learning process, adequate teaching also relies on the respect of teachers for their students. We initiated this research after a conversation with a group of university students, who expressed their discontent regarding the lack of respect shown towards them by some teachers. The results obtained in online surveys highlighted the need for faculty members to analyze and question their attitudes towards their students.

**Abstract:**

The respect of the teacher for the student is essential for effective teaching from the perspective of the students, even in comparison to the knowledge and communication capacity of the teacher. Consequently, the optimal development of this characteristic fosters a more effective and efficient student–teacher relationship. We initiated this research following a conversation with a group of university students, who expressed their discontent regarding the lack of respect shown towards them by some teachers. Therefore, we conducted a descriptive study using online surveys, focusing on the central axis in the teacher–student relationship. The results highlighted the need for faculty members to analyze and question their attitudes towards their students. This paper presents initial results of the data collected at the Veterinary Faculty of the University of Las Palmas de Gran Canaria.

## 1. Introduction

Nowadays, having a university education is one of the more relevant tools enabling considerable changes in society to be made. However, the university environment is very challenging due to the increasing demands, standards, and expectations of the population. Lecturers feel overwhelmed with multiple functions and tasks, such as teaching, conducting promotional research, attending meetings and congresses, requesting projects, participating in community service activities, and helping students in various university activities. Due to these requirements, the characteristics and skills of university teachers are considered to be foremost factors in avoiding negative emotions such as tension, hostility, depression, anger, nervousness, frustration, and even burnout [1,2,3]. Therefore, it is essential for university teachers to possess not only the appropriate knowledge, but also to develop an adequate attitude that would benefit the development of a positive relationship between the student and themselves, as this relationship is one of the fundamental elements in the teaching–learning process [4]. The communication between educators and learners can impact each other either negatively or positively [5]. Hence, teachers must respect the student as a person and be friendly and communicative [6].

The personal characteristics of teachers can help to understand and accept diverse student perspectives, which reinforces the teacher–student relationship [7]. It is known that an effective education process may depend on the teacher’s characteristics [8]. Anderson [9] summarized the characteristics associated with effective teachers, such as confidence, reliability, commitment, analytical and conceptual thinking, information search, initiative, flexibility, responsibility, passion for learning, and respect. This last attribute is particularly valuable, as treating students with respect and expecting the same in return enhances the students’ learning progress [10] and is a motivating factor for students [11].

Delaney et al. [12] analyzed the latter characteristic and concluded that effective teachers had a sense of respect for their students; this was found to be the most essential attribute, even more so than knowledge. Similar results were reported by Bahador et al. and Al-Mohaimeed [13,14]. According to the American philosopher and essayist Ralph Waldo Emerson (1803–1882), “The secret of education lies in respect for the disciple”. However, respect is a controversial concept, and its implications can vary from one culture to another, especially in Western cultures, where respect is considered a mutual and fundamental obligation [15]. It raises the question: how does a professor show respect to their students? Osborne [16] suggested that the common denominator of all answers is the rule of ethical conduct: Do unto others as you would have them do unto you. The relationship between a teacher and a student is more productive when there is mutual respect, and an environment of respect, caring, and trust increases teaching effectiveness [17].

The impetus for this study arose because of the complaints of first-year veterinary college students at the University of Las Palmas de Gran Canaria (ULPGC) regarding disrespectful behavior from some teachers. Thus, this study aimed to assess student perceptions of teacher respect towards students for different academic years within the same faculty.

## 2. Materials and Methods

### 2.1. Study Design and Questionnaire

We conducted a descriptive survey from May to June 2018. The study population (n = 434) comprised students from the Veterinary Faculty of Las Palmas de Gran Canaria University (Canary Islands, Spain), who were in their first to fifth year of study. The inclusion criteria were that students were enrolled in an academic year and had willingness to participate in the study. Answers were collected anonymously via convenience sampling. The minimum sample size was calculated (given a total of 79 answers for a confidence level of 95% and an absolute accepted error of 10%.) using Win Episcope 2.0 [18]. A total of 142 surveys was obtained, and Google Forms (forms.google.com) was used to create and collect answers from the questionnaires. 

Before enabling data collection for the questionnaire, we verified the time required for its completion, and communicated this to the student cohort. The survey link was sent through the Moodle platform corresponding to each relevant academic subject, accompanied by explicit elucidation that participation denoted a deliberate bestowal of informed consent. Prior to answering the questionnaire, the students were provided with an informed consent statement that explained the survey’s purpose, estimated completion time, and assured confidentiality. As the survey targeted the students of the Veterinary Faculty, the response rate could be calculated despite conducting the study using an anonymous online Google Form. 

The researchers of the survey designed a questionnaire to measure student perceptions of respect at the university. It comprised 25 questions grouped into three domains: student demographic profile (gender, date of birth, and year of study), positive attitudes of the teacher, and negative attitudes of the teacher. Table 1 shows questions 1 to 14, which are concerned with the attitude of teachers from a positive point of view, and questions 15 to 22, which are concerned with the attitude of teachers from a negative point of view.

### 2.2. Statistical Analysis

The questionnaire responses were collected in an Excel spreadsheet, and the same program (Excel 2016) was used to perform descriptive statistical analysis and create graphical representations. Data were entered, and statistical analysis was carried out using SPSS 19 (Statistical Package for the Social Sciences). The demographic attributes, including gender, age, and the year of study, were analyzed using descriptive statistical methods. We assessed the rest of the questions using a five-point Likert scale, where students indicated the extent to which they agreed with each statement (1—None, 2—Almost none, 3—About half, 4—The majority, 5—All). Regarding the responses on the Likert scale, in a subsequent analysis, the answers were simplified into two options. Responses of “All” and “Most” relating to positive attitudes were considered as “Most”, while the remaining responses were categorized as “Some”. On the contrary, for responses concerning negative attitudes, the ‘Some’ option included “None” or “Some”, and the rest were considered as “Most”.

The responses were ordinal variables, so descriptive analysis was based on the median, interquartile range, and mode calculation. For inferential analysis, the chi-squared test was used to analyze the gender effect. Somers’ D statistics were calculated to measure the correlation between ordinal variables (Likert scale and academic year), and Cronbach’s alpha was used to assess the reliability and consistency of each domain in the questionnaires. The alpha error was set at 0.05.

## 3. Results

In this study, a total of 142 students completed an online questionnaire. The response rate was 29.89%, and the absolute error of the sample size was 6.75%. Incorrect responses to the birth date question were given by 11 students (7.7%), indicating that their age was missing. The age range was 18–42 years old, with an average age of 22.75, a median of 21, and a mode of 19 years old. Most participants were female (99, 69.72%), with 43 males (30.28%). The distribution of answers related to the academic year in descending order was first year (45%), fifth year (19%), second year (16%), fourth year (11%), and third year (9%). The distribution of answers with taking into account year of study and gender is shown in Figure 1.

Regarding reliability and consistency, we obtained a Cronbach’s alpha of 0.935 for the second domain (positive attitude of the teacher) and 0.865 for the third domain (negative attitude of the teacher). Cronbach’s alpha is a measure of internal consistency, which is closely related to the homogeneity of a set of items. A high value suggests that the items within the scale or questionnaire are strongly correlated with each other, indicating a high degree of interrelatedness and coherence in measuring the targeted construct.

The results of the frequency, percentage, median, interquartile range, and mode of the levels of the Likert scale are presented in Table 2. These questions pertain to two domains: positive attitudes of teachers (Questions 1–14) and negative attitudes of teachers (Questions 15–22). Focusing on the results of the first 14 questions (first domain), we can see that the questions with a median answer of “The majority” were Questions 1 (My teachers treat all the students equally), 3 (My teachers are polite with the students), 8 (My teachers show a receptive and respectful attitude in their relationship with the students), 10 (My teachers respect the students’ diversity) and 11 (The personal behavior I have received from my teachers has been correct). In these questions, more than 85 students (approximately 60% of the sample) reported that “The majority” or “All” of the teachers complied with these premises. In this domain, Question 5 (My teachers motivate me in my studies and professional future) had the worst rating, with 51 students reporting that “None” (7) or “Some” (44) of the teachers motivated the students. Moreover, it is fundamental to highlight that concerning the questions such as whether the teachers tend to smile (Question 2), are willing to help students who have difficulties (Question 7), apologize when they make a mistake (Question 12), have a sense of humor (Question 13) and show enthusiasm (Question 14) (see Table 2), more than 25 students responded with “None” or “Some”.

Figure 2 presents a simplification of the results for the questions regarding positive attitudes of the teachers towards the students, while in Figure 3, we can see the results related to negative attitudes. Both figures present outcomes analogous to those found in Table 2, but they are dichotomized to enhance the interpretation of the results. In Figure 2, we observed that the majority of the teachers treat the students fairly, are polite, are receptive/respectful, respect diversity, and are courteous in their manner, while only a few smile, have a positive attitude, and are patient, motivate, assist students, are approachable, apologize for a mistake, have a sense of humor, or show enthusiasm. Statistical analysis revealed statistically significant disparities across most traits between the two groups.

Table 2 shows the results of questions 15 to 22, which addressed the negative attitudes of the teachers. Five of these questions had a median of “Some” or “None”. At least 75 students (>52% of the sample) considered that “None” or “Some” of the teachers were rude (Question 17), shouted at the students (Question 18), were vengeful (Question 19), humiliated the students (Question 20), or threatened or coerced them (Question 21). The latter question was rejected by nearly 86% of the students. Additionally, 17 students considered that most or all teachers were sarcastic (Question 22), 19 considered that they presented sexist attitudes (Question 16), and 30 reported authority abuse (Question 15). Notable attention should be given to this last result, as more than 21% of the students had a negative perception of the teaching staff. In Figure 3, we observed that behaviors such as being rude, shouting, presenting vengeful attitudes, humiliating, and threatening the students are not commonly observed. However, students feel that a majority of the teachers abuse their authority, exhibit sexist attitudes, and employ sarcasm. Statistically significant differences were found in all of these questions, except for being rude.

Another variable used for stratifying the data was the academic year. Questions corresponding to the positive attitudes of the teachers (7, 9, 11, and 13) were correlated with the academic year using Somers’ D statistic with a value >0.8, and questions 15, 17, and 22 about the negative attitude of teachers had a value >0.7 (Table 2). Somers’ D is a measure of association used to assess the strength and direction of the relationship between two ordinal variables. It ranges from −1 to 1, where 0 indicates no association, positive values suggest a positive association.

## 4. Discussion

The present study investigated the student perceptions of teachers’ respect towards students of different academic years of the Veterinary College of ULPGC. Interestingly, most of the respondents to the survey were in their first year of study. It was unsurprising, as these were the ones who initiated the work and thus became more engaged in the study. Moreover, the fact that more female students than males responded to the questionnaire was due to the increased presence of female students in Spanish universities [19], particularly in veterinary faculties, where feminization is a statistical reality [20], which is in line with other international veterinary faculties [21]. It is particularly pertinent in the ULPGC, where the percentage of female students is very high (72.9%), and a similar percentage was found at our veterinary faculty. However, no association was found between student gender and the other variables, as previously found by Mortazavi et al. [22]. We did not ask if the students were nationals or foreigners because the latter represent less than 3% of the total, and asking this question could have jeopardized their anonymity.

The fact of asking questions about the attitudes of the professors from a positive or negative point of view was conducted to avoid the drawbacks of the Likert scale, since the positive answers always exceeded the negative ones. In our study, we wanted to obtain results more reflective of reality than the manipulation of both types of questions, as recommended by Barnette [23]. Furthermore, by analyzing the consistency of the question within the domains, gave a Cronbach alpha value of 0.935 for the second domain (positive attitude of the teacher) and 0.865 for the third domain (negative attitude of the teacher). According to George and Mallery [24], who provided widely referenced guidelines for interpreting Cronbach’s alpha, the values above 0.9 provide excellent internal consistency, while values between 0.8 and 0.9 show good internal consistency. Therefore, a Cronbach’s alpha of 0.935 and 0.865 falls within the “excellent and good internal consistency” range, indicating that the items in the scale are reliably measuring the same underlying construct. This level of internal consistency lends support to the validity and reliability of the measurement instrument [25]. This result was in contrast with other authors who pointed out issues such as internal consistency when negatively worded questions were used [23,25,26]. 

Regarding the academic year, Somers’ D analysis showed that appreciation for the teacher’s attitude increased only in seven of the 22 questions. However, we must consider that questions 7, 9, 11, and 13 are included in the domain of positive teacher attitude, while questions 15, 17, and 22 deal with the negative teacher attitude (Table 1). Interpreting a Somers’ D value higher than 0.7 indicates a strong positive association between two ordinal variables and suggests a substantial degree of correlation between the two variables, implying that as one variable increases, the other tends to increase as well [27].

In this study, the results of the Likert scale questions were presented in two ways: the complete information provided in Table 2, and the dichotomized results depicted in Figure 2 and Figure 3. Dichotomizing the results of a five-degree Likert scale offers potential benefits, such as simplifying the analysis, facilitating communication, highlighting extreme values, allowing direct comparison of proportions in the two groups. However, it is essential to note that dichotomizing Likert scale data also entails limitations, including information loss, misinterpretation, and oversimplification of participant responses [25].

Regarding the positive attitudes expected from the teachers, it seems that most of the students found that most of the teachers treat them in a correct manner. Question 10 inquired about whether the teachers respect student diversity. Nearly 70% of the students responded that most of the teachers do so. At the veterinary faculty of ULPGC, diversity is more closely linked to gender diversity, given the homogeneity of the student body. In fact, ULPGC promotes initiatives for raising awareness and providing training, along with measures to support research, aimed at the university community to promote respect and ensure the protection of the right to freely self-determine gender identity and expression. Cornell et al. [28] studied the significance of inclusive learning environments and found that respectful teacher–student relationships positively affected students from diverse backgrounds. These relationships created a sense of belonging and engagement among students, ultimately contributing to improved academic performance and retention. The question most assessed by the students was Question 11, which referred to the correct treatment received from them, with 102 out of 142 (71.8%) agreeing.

In the modern educational landscape, the role of university teachers extends beyond the mere transmission of knowledge. Teachers are entrusted with the crucial task of creating an inclusive environment that ensures effective and engaging instruction. However, a paradox emerges when some educators succeed in cultivating an inclusive atmosphere yet fall short in delivering captivating and effective teaching methodologies. In this study, we detected a failure in most of the teachers in the affective–motivational dimension, which refers to the emotional and motivational aspects of the teaching and learning process. It encompasses the teacher’s ability to create a positive emotional climate, foster motivation, and establish a supportive relationship with students. It recognizes that emotions, attitudes, and motivation play a significant role in shaping students’ learning experiences and outcomes. Thus, Question 5, in which students felt that the teachers did not motivate them in their studies and professional future, was the least rated, with only 38 out of 142 (26.8%) indicating that teachers motivated them. 

While the promotion of a positive and inclusive environment is essential for nurturing a supportive learning community, it is equally crucial for teachers to complement this effort with engaging and effective teaching practices. By recognizing the divergence in skill sets, addressing pedagogical barriers, and embracing a willingness to evolve, educators can bridge the gap between inclusivity and engagement. Through a balanced approach, teachers can empower their students with not only a sense of belonging but also a passion for learning that extends far beyond the classroom. Ryan and Deci’s self-determination theory [29] emphasizes the importance of fostering intrinsic motivation and autonomy in students’ learning, arguing that respectful and supportive teacher–student relationships satisfy students’ psychological needs, promoting intrinsic motivation and active engagement in the learning process. 

Previous evidence indicates that students believe that a positive teacher–student relationship is essential to improve the educational environment [30]. Other studies reported that the quality of the teacher–student relationship has a strong impact on learning, performance, and students’ satisfaction [31]. Furthermore, the teachers’ behavior has a notable role in the teacher–student relationship, while other variables such as gender, age, and physical appearance of the teacher have lesser influence [3,22,32]. Another characteristic of outstanding teachers is the demonstration of enthusiasm in their classroom, despite having taught the same subject for many years [17]. Delaney et al. [12] analyzed positive attitudes of educators and considered that effective teachers had a sense of respect for their students. This is because students were more likely to consider teachers who were compassionate and understanding, treated them with respect, and made them feel comfortable asking questions. The same authors asked their students what characteristics were essential for effective teaching from the student’s perspective, and respect turned out to be the most important, even more so than knowledge and the ability to communicate. In addition, we should consider that a sense of humor in university teaching could play a significant role in creating a positive and engaging learning environment by promoting a sense of warmth and approachability. Humor has the potential to reduce stress and anxiety among students. This reduction can enhance cognitive functioning and information retention and help to capture students’ attention and maintain their engagement with the material [33].

The relationship between educators and students is a critical factor that significantly influences students’ academic performance, emotional well-being, and overall learning experience. Numerous studies emphasize the role of positive teacher–student relationships in academic achievement. Roorda et al. [34] conducted a meta-analysis involving 99 studies and found that supportive teacher–student relationships positively correlated with higher academic motivation and engagement. While a longitudinal study by Hamre and Pianta [35] revealed that positive teacher–student interactions were associated with greater academic progress over time. Effective communication is a crucial component of a respectful connection between educators and students. Martin and Dowson [36] investigated the role of communication in teacher–student relationships and highlighted that open and transparent communication enhances the quality of the relationship and student satisfaction with the learning experience. Also, the emotional well-being of students is deeply influenced by their relationship with educators. Jennings and Greenberg [37] explored the impact of teacher–student relationships on student mental health and found that supportive relationships contributed to reduced stress and anxiety levels. Conversely, negative relationships were associated with emotional distress and academic disengagement.

Concerning the negative attitudes of the teachers observed in our study, thirty out of 142 students (21.1%) reported that most of the teachers had abused their authority, whereas 82 (57.7%) answered that about half or more had exhibited sexist attitudes or been sarcastic. Gender bias is a prevalent issue within the medical field, often taking the form of microaggressions that begin to surface during medical school. This study highlights that female medical students consistently experience significant microaggressions, resulting in heightened stress levels. Among the various microaggression domains, the concept of “leaving gender at the door” emerges as the predominant and particularly distressing category. It may potentially signify the existence of societal pressure to downplay feminine attributes in the pursuit of success. [38]. In the context of veterinarians practicing in rural areas, it has been observed that the prevalence of sexist attitudes among certain farmers and colleagues constitutes a significant disadvantage [39]. Sometimes, sarcasm is considered a sort of offensive humor, but it is considered inappropriate in the classroom. Research has shown that the use of sarcasm by educators can have negative consequences. Sarcasm, when not well-received, can lead to misunderstandings, create a hostile classroom environment, and hinder effective communication. It might impede students’ comprehension, discourage participation, and adversely affect teacher–student relationships [40]. 

Fortunately, most students did not feel threatened/coerced by the teachers (122/142, 85.9%). Additionally, most of them thought that most teachers did not shout at them (105/142, 73.9%) nor display a vengeful attitude (101/142, 71.1%). Different authors focus on the teacher’s respect towards students in order to maintain an appropriate learning environment, while others agree that mutual respect is essential for student–teacher communication [17,41,42,43]. Therefore, avoiding negative attitudes and constructing a respectful association and interaction between students and teachers is essential to stimulate and encourage the learning of a subject, as the impact of educator relational behaviors on educational responsibility, favorable outcome, and enthusiasm of beginners is the basis of most studies [44,45]. Furthermore, a quality teacher–student relationship has been linked to more enjoyment and less anxiety and anger [46]. It is important to remember that teaching is not only about imparting knowledge; it should also concern the development of an interest and love for learning. Teachers must inspire and motivate students [47].

## 5. Conclusions

The results of this study emphasize the importance of nurturing a respectful connection between educators and students. Positive teacher–student relationships have been linked to improved academic achievement, enhanced emotional well-being, increased intrinsic motivation, and the creation of an inclusive learning environment. Therefore, educational institutions should implement strategies and initiatives to promote a culture of respect and support, ultimately to enrich the educational experience for all students. To evaluate further this perspective, studies involving more veterinary colleges should be conducted. In addition, although the teachers received positive assessments across most surveyed aspects, the absence of affective–motivational dimensions towards students warrants an active review. These attitudes can impact students’ development and future professional endeavors. Similarly, the presence of sexist attitudes and instances of authority abuse identified through this survey should be critically examined and controlled to eliminate such behavior toward students within the university.

## Figures and Tables

**Figure 1 vetsci-10-00538-f001:**
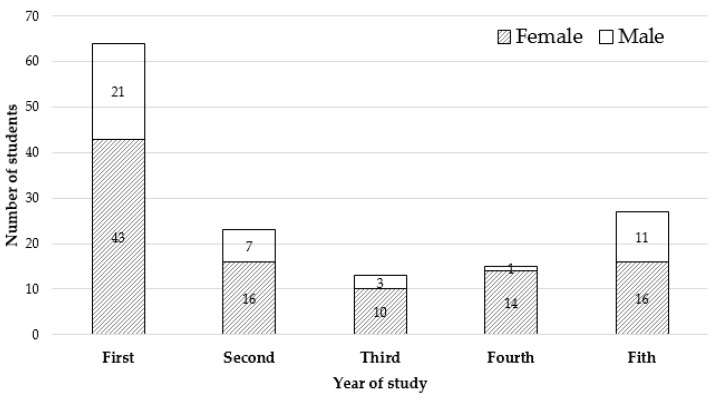
Distribution and gender of the students related to the year of study.

**Figure 2 vetsci-10-00538-f002:**
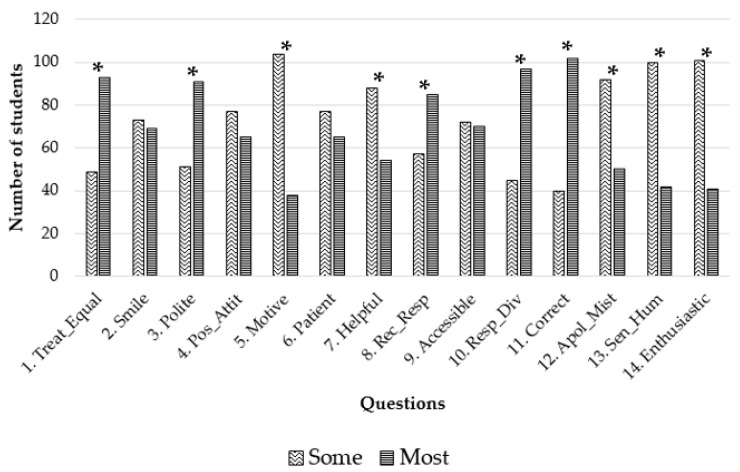
Dichotomized answers to the questions about positive attitudes of the teachers. Statistically significant differences in proportions are marked with an asterisk.

**Figure 3 vetsci-10-00538-f003:**
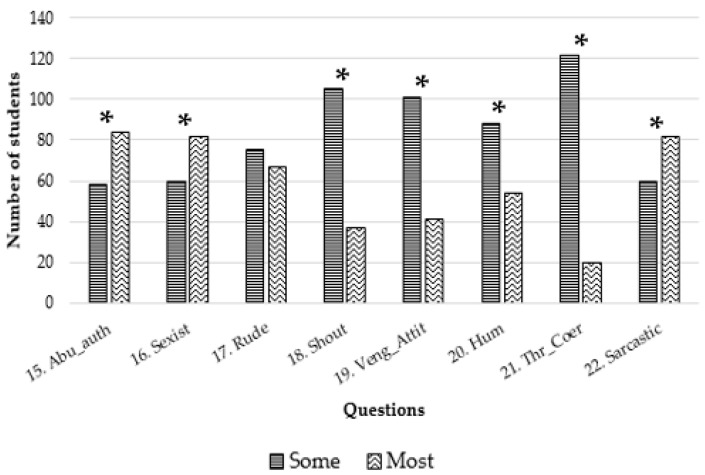
Dichotomized answers to the questions about negative attitudes of the teachers. Statistically significant differences in proportions are marked with an asterisk.

**Table 1 vetsci-10-00538-t001:** Questionnaire used to measure student perceptions of respect at the university. Questions in Spanish, the original language of the study, are shown in parenthesis and in grey.

Domain	Question	Abbreviation
Positive attitudes of the teachers	1. My teachers treat all the students equally (Mis profesores tratan a todos los alumnos por igual)	Treat_Equal
	2. My teachers usually smile (Mis profesores suelen sonreír)	Smile
	3. My teachers are polite with the students (Mis profesores son educados con los estudiantes)	Polite
	4. My teachers have a positive attitude towards the students (Mis profesores tienen actitud positiva hacia los alumnos)	Pos_Attit
	5. My teachers motivate me in my studies and professional future (Mis profesores me motivan en mis estudios y futuro profesional)	Motive
	6. My teachers have patience (Mis profesores tienen paciencia)	Patient
	7. My teachers are willing to help students who have difficulties (Mis profesores se muestran dispuestos a ayudar a los estudiantes que tienen dificultades)	Helpful
	8. My teachers show a receptive and respectful attitude in their relationship with the students (Mis profesores manifiestan una actitud receptiva y respetuosa en su relación con el alumnado)	Rec_Resp
	9. My teachers are accessible to the students (Mis profesores se muestran accesibles a los estudiantes)	Accessible
	10. My teachers respect the student diversity (Mis profesores respetan la diversidad)	Resp_Div
	11. The personal behavior I have received from my teachers has been correct (El trato personal que he recibido por parte de mis profesores ha sido correcto)	Correct
	12. My teachers apologize when they make a mistake (Mis profesores se disculpan cuando cometen un error)	Apol_Mist
	13. My teachers have a sense of humor (Mis profesores tienen sentido del humor)	Sen_Hum
	14. My teachers show enthusiasm (Mis profesores muestran entusiasmo)	Enthusiastic
Negative attitudes of the teachers	15. My teachers abuse their authority (Mis profesores abusan de su autoridad)	Abu_auth
	16. My teachers present sexist attitudes (Mis profesores presentan actitudes sexistas)	Sexist
	17. My teachers are rude to the students (Mis profesores son maleducados con los alumnos)	Rude
	18. My teachers shout at the students (Mis profesores gritan a los alumnos)	Shout
	19. My teachers present a vengeful attitude toward the students (Mis profesores presentan una actitud vengativa hacia los alumnos)	Veng_Attit
	20. My teachers humiliate the students (Mis profesores humillan a los alumnos)	Hum
	21. My teachers threaten/coerce the students (Mis profesores amenazan/coaccionan a los alumnos)	Thr_Coer
	22. My teachers are sarcastic when dealing with the students (Mis profesores son sarcásticos en el trato con los alumnos)	Sarcastic

**Table 2 vetsci-10-00538-t002:** Descriptive statistics, the responses for the five levels of the Likert scale (1–None 2–Some 3–About half 4–The majority 5–All) are expressed as frequencies and percentages of total student responses.

Domain	Question ^1^	None	Some	About Half	The Majority	All	Median (Q3–Q1 ^#^)	Mode	Somers’ d
**Positive attitudes of the teachers**	1. Treat_Equal	2 1.41%	11 7.75%	36 25.35%	80 56.34%	13 9.15%	The majority 1	The majority	0.545
	2. Smile	1 0.7%	27 19.01%	45 31.69%	62 43.66%	7 4.93%	About half 1	The majority	0.059
	3. Polite	0 0.00%	11 7.75%	40 28.17%	78 54.93%	13 54.93%	The majority 1	The majority	0.169
	4. Pos_Attit	0 0.00%	22 15.49%	55 38.73%	59 41.55%	6 4.23%	About half 1	The majority	0.345
	5. Motivate	7 4.93%	44 30.99%	53 37.32%	33 23.24%	5 3.52%	About half 2	About half	0.481
	6. Patient	2 1.41%	22 15.49%	53 37.32%	58 40.85%	7 4.93%	About half 1	The majority	0.558
	7. Helpful	3 2.11%	31 21.83%	54 38.03%	45 31.69%	9 6.34%	About half 1	About half	0.909
	8. Rec_Resp	1 0.7%	15 10.56%	41 28.87%	78 54.93%	7 4.93%	The majority 1	The majority	0.082
	9. Accessible	1 0.7%	17 11.97%	54 38.03%	61 42.96%	9 6.34%	About half 1	The majority	0.900
	10. Resp_Div	0 0.00%	12 8.45%	33 23.24%	58 40.85%	39 27.46%	The majority 2	The majority	0.135
	11. Correct	0 0.00%	4 2.82%	36 25.35%	75 52.82%	27 19.01%	The majority 1	The majority	0.950
	12. Apol_Mist	11 7.75%	34 23.94%	47 33.1%	37 26.06%	13 9.15%	About half 2	About half	0.208
	13. Sen_hum	1 0.7%	25 17.61%	74 52.11%	35 24.65%	7 4.93%	About half 1	About half	0.811
	14. Enthusiastic	3 2.11%	35 24.65%	63 44.37%	38 26.76%	3 2.11%	About half 2	About half	0.386
**Negative attitudes of the teachers**	15. Abu_auth	21 14.79%	37 26.06%	54 38.03%	29 20.42%	1 0.7%	About half 1	About half	0.740
	16. Sexist	30 21.13%	30 21.13%	63 44.37%	18 12.68%	1 0.7%	About half 1	About half	0.276
	17. Rude	26 18.31%	49 34.51%	60 42.25%	7 4.93%	0 0.00%	Some 1	About half	0.728
	18. Shout	52 36.62%	53 37.32%	37 26.06%	0 0.00%	0 0.00%	Some 2	Some	0.538
	19. Veng_Attit	36 25.35%	65 45.77%	37 26.06%	4 2.82%	0 0.00%	Some 2	Some	0.005
	20. Hum	22 15.49%	66 46.48%	52 36.62%	1 0.7%	1 0.7%	Some 1	Some	0.572
	21. Thr_Coer	90 63.38%	32 22.54%	19 13.38%	0 0%	1 0.7%	None 1	None	0.547
	22. Sarcastic	18 12.68%	42 29.58%	65 45.77%	15 10.56%	2 1.41%	About half 1	About half	0.708

^#^ Quartile 3 minus quartile 1 in Likert scale. ^1^ Abbreviations are shown in Table 1.

## Data Availability

Not applicable.

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
