# Peer review of "Nurturing a Respectful Connection: Exploring the Relationship between University Educators and Students in a Spanish Veterinary Faculty"

_vetsci, 2023, doi:10.3390/vetsci10090538_

Round 1
Reviewer 1 Report
This paper addresses the important topic of student-teacher relationships, using the example of undergraduate students’ experience in the veterinary faculty of Las Palmas de Gran Canaria University. While the data are interesting as a case study, with the authors concluding that ‘a climate of respect in the classroom is vital for students,’ I wasn’t convinced that strong conclusions could be drawn from the data. Points for the authors to consider in a revision include:
1. Consider revising the title by adding: ‘... in a Spanish veterinary faculty.’ That defines the scope.
2. The conclusions don’t match between the abstract and the conclusions section of the paper. The abstract says that ‘The results highlighted the need for faculty members to ... their students’ (lines 27-29), while the conclusions emphasize ‘a climate of respect’ (line 225) and ‘Respect is always paramount’ (line 237). I don’t see how ‘...question(ing) their attitude to their students’ automatically implies respect.
3. I assume that the original questionnaire was given in Spanish, with the items in the paper translated into English. It will be helpful to have the original questionnaire, in Spanish, available as supporting information. That way any nuances of translation can be checked by readers if they wish.
4. I didn’t understand what is meant by ‘My teachers are educated with the students’ in Table 1. Does it mean that students and teachers learn from each other in the classroom? This is likely a simple issue with translation, where a literal translation misses an idiomatic quality. It should be clarified.
5. Lines 99-100 – Just leave this sentence as: ‘An alpha error was set at 0.05.’ The rest of the statement is redundant.
6. In the results, it would be useful to know a little more about the cohort from which the sample was drawn and how the sample compared to the cohort: are the sex ratio/age distribution/academic distribution in the sample the same as in the cohort? Finally, is the cohort ethically uniform (i.e., Spanish students), or is there an international element? If there is an international element, might they have a different perspective? Given that one of the questions relates to student diversity (Table 1, point 10), there is an implication that there is a diversity in the cohort beyond gender, age or year level.
7. Lines 175-176 – change ‘as recommended by an author’ to ‘as recommended by Barnette’ to match the approach used in other citations.
8. Given the descriptive nature of the study, it is hard to draw strong conclusions. For example, might students who had been disciplined be more likely to comment negatively on staff behaviour? How can one conclude that respect is always paramount when the strongest negative impression was failure to motivate students? It may be best to describe the results, offer reflections on what it means for the faculty under study, and not push the data further.
If I was as fluent in Spanish as the authors are in English, I'd be pleased. That said, there are occasional infelicities of expression in the paper that could be improved by a native English speaker.
Author Response
Dear Reviewer,
We appreciate your suggestion and comments, which have improved our manuscript.
Points that have been considered in the revised version include:
- Consider revising the title by adding: ‘... in a Spanish veterinary faculty.’ That defines the scope. "As you recommend, we have added this sentence to the title".
- The conclusions don’t match between the abstract and the conclusions section of the paper. The abstract says that ‘The results highlighted the need for faculty members to ... their students’ (lines 27-29), while the conclusions emphasize ‘a climate of respect’ (line 225) and ‘Respect is always paramount’ (line 237). I don’t see how ‘...question(ing) their attitude to their students’ automatically implies respect. "Following your suggestion, we have redone the conclusion section to better match the abstract."
- I assume the original questionnaire was given in Spanish, with the items in the paper translated into English. It will be helpful to have the original questionnaire, in Spanish, available as supporting information. That way any nuances of translation can be checked by readers if they wish. "Following your suggestion, we have included the Spanish questionnaire, after each English question."
- I didn’t understand what is meant by ‘My teachers are educated with the students’ in Table 1. Does it mean that students and teachers learn from each other in the classroom? This is likely a simple issue with translation, where a literal translation misses an idiomatic quality. It should be clarified.
"Thank you for the comment. We have rewritten the question as follows: My teachers are polite with the students."
- Lines 99-100 – Just leave this sentence as: ‘An alpha error was set at 0.05.’ The rest of the statement is redundant. "We have redone the sentence as you suggested."
- In the results, it would be useful to know a little more about the cohort from which the sample was drawn and how the sample compared to the cohort: are the sex ratio/age distribution/academic distribution in the sample the same as in the cohort? Finally, is the cohort ethically uniform (i.e., Spanish students), or is there an international element? If there is an international element, might they have a different perspective? Given that one of the questions relates to student diversity (Table 1, point 10), there is an implication that there is a diversity in the cohort beyond gender, age or year level.
"The target population present a sex ratio similar to the sample one. And this information has been added as: “This is particularly pertinent in the ULPGC, where the percentage of female students is very high (72.9%) and a similar percentage is found at our veterinary faculty."
"Related to foreign students they just represent less than 3%. This was a variable that we did not ask in order to preserve the anonymity of the students. We have added the following information: “We did not ask if the students were nationals or foreigners, because the latter represents less than 3% of the total, and asking this question could jeopardize their anonymity.”
"Information about student diversity has been added in L 265-270: At the veterinary faculty of ULPGC, diversity is more closely linked to gender diversity, given the homogeneity of the student body. In fact, ULPGC promotes initiatives for raising awareness and providing training, along with measures to support research, aimed at the university community to promote respect and ensure the protection of the right to freely self-determine gender identity and expression.”
-
Lines 175-176 – change ‘as recommended by an author’ to ‘as recommended by Barnette’ to match the approach used in other citations. "We have changed these sentences following your recommendation".
- Given the descriptive nature of the study, it is hard to draw strong conclusions. For example, might students who had been disciplined be more likely to comment negatively on staff behaviour? How can one conclude that respect is always paramount when the strongest negative impression was the failure to motivate students? It may be best to describe the results, offer reflections on what it means for the faculty under study, and not push the data further.
"We strongly agree with your comment. Therefore, the conclusion has been rewritten as mentioned before"
Reviewer 2 Report
Ramirez et al. conducted a study that surveyed veterinary students at the University of Las Palmas de Gran Canaria (Spain) to explore various aspects of teachers' respect towards students.
The work is well presented, reads well and is clearly followed in its formulation, objectives and results.
This subject has received limited attention in veterinary medicine faculties, making the study novel and potentially interesting for other educational institutions. However, the study's localized nature restricts the generalizability of the findings, offering only a snapshot of that specific university.
The survey employed Google Forms and collected responses from nearly a third of the students. The authors mention using Win Episcope 2.0 to calculate the minimum sample size, but they don't provide the results. It raises the question of whether the number of students responding, particularly when the sample is divided into five grades, is sufficient to draw reliable conclusions. For instance, in the 3rd grade, the response count should be around 12-13 students.
One must question if this number allows for drawing conclusions regarding the variation in students' perception of their teachers' attitudes over the academic years. This is an interesting point. It would be pertinent to investigate whether students' opinions change as they progress through their academic careers. While the faculty members teaching the courses may vary, understanding if students' overall feeling of respect towards their professors is similar or different as they advance in their academic journey would be insightful. Although the study indicates an increased appreciation in the students' appreciation increased in relation to the year in 7 questions, but 3 of them were negative, making it difficult to conclude a definitive increase in appreciation. This aspect requires clarification.
In conclusion, despite the limitations in methodology and conclusions, this study opens up an intriguing avenue for exploring teacher-student relations at the university level.
Minor concerns:
- In Material and Methods, it is stated that the question was about “gender” of the students. Consequently, whether or not there are differences in the responses will be by “gender” and not by “sex” (line 154).
- in lines 165-168 they justify the majority female presence in the ULPG in line with what occurs in Spanish universities in general, providing a reference [19] from 2002. It would be convenient to provide a more recent reference on the subject.
- Is the reference [12] complete. It seems that the publication data are missing.
Author Response
Dear Reviewer,
First of all, we appreciate all your comments since these have improved the manuscript. We have divided our responses in:
Major comments
- The survey employed Google Forms and collected responses from nearly a third of the students. The authors mention using Win Episcope 2.0 to calculate the minimum sample size, but they don't provide the results. It raises the question of whether the number of students responding, particularly when the sample is divided into five grades, is sufficient to draw reliable conclusions. For instance, in the 3rd grade, the response count should be around 12-13 students.
Thank you for the comment. The following information has been added in lines 83-84: “…, given a total of 79 answers for a confidence level of 95% and an absolute accepted error of 10%.”
Related to the analysis of the relationship between the year of study and the answers. We used the Somers’ D test, which is a measure of association used to assess the strength and direction of the relationship between two ordinal variables. It ranges from -1 to 1, where 0 indicates no association, positive values suggest a positive association and negative values suggest a negative association. In seven of the questions, the Somers’ D gave values higher than 0.7, three of them being higher than 0.9. This shows a tendency in our results and a strong relationship.
Therefore, We have added the following sentence to the text: “Interpreting a Somers' D values higher than 0.7 indicates a strong positive association between two ordinal variables and suggests a substantial degree of correlation between the two variables, implying that as one variable increases, the other tends to increase.”
- One must question if this number allows for drawing conclusions regarding the variation in students' perceptions of their teachers' attitudes over the academic years. This is an interesting point. It would be pertinent to investigate whether students' opinions change as they progress through their academic careers. While the faculty members teaching the courses may vary, understanding if students' overall feeling of respect towards their professors is similar or different as they advance in their academic journey would be insightful. Although the study indicates an increased appreciation in the student's appreciation increased in relation to the year in 7 questions, 3 of them were negative, making it difficult to conclude a definitive increase in appreciation. This aspect requires clarification.
We agree. It would be interesting to investigate whether students' opinions change as they progress through their academic careers. But so far, with the answers that we have, it is what we could conclude. Based on your comment and another from the other reviewer, the conclusion has been rewritten as: “Scientific research consistently emphasizes the importance of nurturing a respectful connection between university educators and students. Positive teacher-student relationships have been linked to improved academic achievement, enhanced emotional well-being, increased intrinsic motivation, and the creation of an inclusive learning environment. Effective communication plays a vital role in fostering these relationships. As educational institutions recognize the significance of these connections, they can implement strategies and initiatives to promote a culture of respect and support, ultimately enriching the educational experience for all students. To further evaluate this perspective, studies involving more veterinary colleges should be conducted. Furthermore, while the teaching staff has received positive assessments across most surveyed aspects, the absence of affective-motivational dimensions toward students warrants an active review. These attitudes can directly impact students' development and future professional endeavours. Similarly, the presence of sexist attitudes and instances of authority abuse identified through this survey should be critically examined and controlled to eliminate such behaviour toward students within the university and broader society.”
Minor comments
- In Material and Methods, it is stated that the question was about the “gender” of the students. Consequently, whether or not there are differences in the responses will be by “gender” and not by “sex” (line 154).
As you recommend, we have changed it.
- in lines 165-168 they justify the majority female presence in the ULPG in line with what occurs in Spanish universities in general, providing a reference [19] from 2002. It would be convenient to provide a more recent reference on the subject.
Done. A new reference has substituted the old one: "Gómez Marcos, M.T.; Vicente Galindo, M.P.; Martín Rodero, H. Mujeres en la universidad española: diferencias de género en el alumnado de grado. Revista INFAD De Psicología. International Journal of Developmental and Educational Psychology 2019, 2, 443–454. https://doi.org/10.17060/ijodaep.2019.n1.v2.1484".
- Is the reference [12] complete? It seems that the publication data are missing.
You are right. Therefore, the complete reference has been added:
Delaney, J.G.; Johnson, A.; Johnson, T.D.; Treslan, D. Students’ Perceptions of Effective Teaching in Higher Education. Project Report. Memorial University of Newfoundland, St. John's, Newfoundland. 2010, https://research.library.mun.ca/8370/.
Round 2
Reviewer 1 Report
Thank you for addressing my comments.
English is fine.